# Cell Type-Specific Role of RNA Nuclease SMG6 in Neurogenesis

**DOI:** 10.3390/cells10123365

**Published:** 2021-11-30

**Authors:** Gabriela Maria Guerra, Doreen May, Torsten Kroll, Philipp Koch, Marco Groth, Zhao-Qi Wang, Tang-Liang Li, Paulius Grigaravičius

**Affiliations:** 1Leibniz Institute on Aging—Fritz Lipmann Institute (FLI), Beutenbergstr 11, 07745 Jena, Germany; gabrielagrra@gmail.com (G.M.G.); Doreen.May@leibniz-fli.de (D.M.); Torsten.Kroll@leibniz-fli.de (T.K.); Philipp.Koch@leibniz-fli.de (P.K.); Marco.Groth@leibniz-fli.de (M.G.); li.tangliang@sdu.edu.cn (T.-L.L.); 2Faculty of Biological Sciences, Friedrich-Schiller University of Jena, Beutenbergstr 11, 07745 Jena, Germany; 3State Key Laboratory of Microbial Technology, Shandong University, No. 72, Binhai Road, Qingdao 266237, China

**Keywords:** SMG6, NMD, neurogenesis, neurodevelopmental syndromes

## Abstract

SMG6 is an endonuclease, which cleaves mRNAs during nonsense-mediated mRNA decay (NMD), thereby regulating gene expression and controling mRNA quality. SMG6 has been shown as a differentiation license factor of totipotent embryonic stem cells. To investigate whether it controls the differentiation of lineage-specific pluripotent progenitor cells, we inactivated *Smg6* in murine embryonic neural stem cells. Nestin-Cre-mediated deletion of *Smg6* in mouse neuroprogenitor cells (NPCs) caused perinatal lethality. Mutant mice brains showed normal structure at E14.5 but great reduction of the cortical NPCs and late-born cortical neurons during later stages of neurogenesis (i.e., E18.5). *Smg6* inactivation led to dramatic cell death in ganglionic eminence (GE) and a reduction of interneurons at E14.5. Interestingly, neurosphere assays showed self-renewal defects specifically in interneuron progenitors but not in cortical NPCs. RT-qPCR analysis revealed that the interneuron differentiation regulators *Dlx1* and *Dlx2* were reduced after *Smg6* deletion. Intriguingly, when *Smg6* was deleted specifically in cortical and hippocampal progenitors, the mutant mice were viable and showed normal size and architecture of the cortex at E18.5. Thus, SMG6 regulates cell fate in a cell type-specific manner and is more important for neuroprogenitors originating from the GE than for progenitors from the cortex.

## 1. Introduction

Cell fate relies on the correct “read” of the genetic code and its translation into functional proteins. Nonsense-mediated mRNA decay (NMD) is a cellular surveillance mechanism that is involved in controlling the quality of mRNA [1,2,3]. It degrades transcripts that, after a nonsense mutation or alternative splicing events, harbor a premature termination codon (PTC) before (>50–55 nucleotides) an exon–exon junction complex (EJC). The stable interaction of UPF1 with eRFs at the PTC site recruits the NMD factors UPF2, UPF3 and the kinase SMG1. SMG1 phosphorylates UPF1 and UPF2 thereby promoting the recruitment of the endonuclease SMG6 or the SMG5/SMG7-mediated exonuclease for RNA degradation [1]. The branches of SMG6- and SMG5/7-mediated NMD pathways have been shown to overlap, yet with distinct differences in certain populations of target transcripts [4]. Recently, it was demonstrated that SMG6 (Suppressor with morphogenetic effect on genitalia protein 6) endonuclease activity can depend on the SMG5/SMG7 heterodimer [5]. Moreover, non-mutant transcripts can also be targeted by NMD, for example the transcripts with uORF and long 3′UTR, thereby regulating normal gene expression [4,6].

NMD deficiency leads to an accumulation of deleterious mRNA products, which can be translated not only into mutated malfunctional proteins but also into functional ones, which then may cause tissue dysfunction or pathogenesis [3,7,8,9]. The complete knockout of NMD genes, such as *Upf1* [10], *Upf2* [11] or *Smg1* [12] resulted in embryonic lethality, highlighting the importance of NMD for early development. Previously, we have shown that the complete deletion of *Smg6* in mouse germline blocks differentiation of embryonic stem (ES) cells into germ layers and thereby results in early embryonic lethality [13]. Furthermore, we demonstrated that SMG6 is required for the production of induced pluripotent stem cells (iPS) from somatic cells [13]. This identified SMG6-mediated NMD as a fundamental regulator of cell fate change, thereby controlling differentiation and development. In that study, the telomeric function of SMG6, originally identified as *Est1a* (Ever Shorter Telomere 1a) in yeast, was proven to be negligible.

In humans, mutations or deletion of NMD factors have been associated with several neurological disorders [14]. Human UPF3B, encoded by the X-linked gene *UPF3B*, was the first NMD factor linked with human neurodevelopmental syndromes such as X-linked intellectual disability (ID) with and without autism, childhood onset schizophrenia (COS) and attention deficit hyperactivity disorder (ADHD) [15,16]. Mutations of *UPF3B* led to the loss or truncation of the protein and caused ID in males of several families [15,16]. Genetic studies have identified mutations or copy number variants in *UPF2*, *UPF3*, *SMG9* and the exon-junction complex component *RBM8A* in human patients with neurological symptoms [17,18,19]. Furthermore, NMD proteins (UPF3b, UPF1, UPF2, SMG1) have been shown to participate in axon outgrowth, synapse formation and, thus, in various behavioral processes [20,21,22,23,24,25,26]. All these observations suggest that, in addition to its general role in the early embryonic development, NMD has specific functions in self-renewal and differentiation of neuroprogenitor cells as well as in neuronal functionality. However, the precise molecular mechanism underlying these processes remains largely unknown.

In order to elucidate the function of SMG6 in neurogenesis during brain development, we applied genetic and cellular studies to inactivate *Smg6* in various neural progenitor cells. We found that SMG6 is more critical for the cell fate determination of interneuron progenitors in the ganglionic eminence (GE) compared to cortical neuroprogenitor cells (NPCs).

## 2. Material and Methods

### 2.1. Mice

Smg6^flox^(*Smg6^tm1.1Zqw^*), Smg6^+/Δ^(*Smg6^tm1.2Zqw^*) [13], Nes-Cre (*Tg*(*Nes-cre*)*1Kln*) [27], Emx1-Cre (*Emx1*^*tm1*(*cre*)*Krj*^) [28], Rosa26-CreER^T2^ (Gt(ROSA)26Sor^tm1(cre/ERT2)Tyj^), *Smg6*-CNSΔ and *Smg6*-CoHi∆ mice were bred and housed in the mouse facility of Fritz Lipmann Institute (FLI, Jena, Germany). Mice were fed ad libitum with standard laboratory chow and water in ventilated cages under a 12 h light/dark cycle. All animal work was conducted according to the German animal welfare legislation and approved by the Thüringer Landesamt für Verbraucherschutz (TLV). The genotyping of mice was performed by PCR on DNA extracted from tail tissue as previously described [13].

### 2.2. Immunoblot Analysis

Proteins were extracted using RIPA buffer supplemented with 1 mM PMSF and 20–50 µg of cell lysates were separated with SDS–PAGE as described [29]. The primary antibodies rabbit anti-*Smg6*/Est1A (1:1500; Abcam, Berlin, Germany), mouse anti-GAPDH (1:20,000; Sigma-Aldrich, Taufkirchen, Germany), mouse anti-Actin (1:20,000; Sigma-Aldrich), and secondary antibodies HRP-conjugated goat anti-rabbit IgG or goat anti-mouse IgG (1:10,000; DAKO, Frankfurt am Main, Germany), were used.

### 2.3. RT-qPCR

Total RNA was isolated from neural stem cells using the RNeasy Mini Kit (Qiagen, Hilden, Germany) following the manufacture’s recommendations. After genomic DNA removal by DNAse I, the cDNA library was generated using SuperScript™ III Reverse Transcriptase (Invitrogen, Thermo Fisher Scientific, Waltham, MA, USA). The quantitative real-time PCR (qPCR) was performed in triplicates for each sample using Platinum™ SYBR™ Green qPCR SuperMix-UDG (Invitrogen) and a LightCycler^®^ 480 Instrument (Roche). The sequences of the used primers are summarized in Table 1.

### 2.4. Neurosphere Formation Assay and In Vitro Neuronal Stem Cell Differentiation

Neuroprogenitor cells were isolated and used for neurosphere formation assays from E13.5 cortices and ganglionic eminences as previously described [29]. After isolation, neuroprogenitor cells were plated for neurosphere formation in DMEM/F12 medium (Gibco, Thermo Fisher Scientific, Waltham, MA, USA) supplemented with 2% B-27 supplement (Invitrogen), 1X penicillin and streptomycin (Thermo Fischer Scientific, Waltham, MA, USA), 20 ng/mL EGF and 10 ng/mL bFGF (PeproTech, East Windsor, NJ, USA). Formed neurosphere numbers and cell numbers were counted after 7 days in culture. For the in vitro differentiation, neurospheres were trypsinised and cells were plated on poly-L-lysine (10 µg/mL overnight at room temperature, P5899, Sigma-Aldrich) and then laminin (10 µg/mL, 30 min at 37 °C, L2020, Sigma-Aldrich)-coated flat bottom 96-well plates (CellCarrier 96 Ultra, Cat# 6005550). After 2 days, the differentiation was initiated by changing the neural stem cell medium by differentiation medium: DMEM/F12 (Gibco) supplemented with 1% FSC (Thermo Fischer Scientific), 2% B27 (Invitrogen), 1X penicillin and streptomycin (Thermo Fischer Scientific). Cells were fixed, permeabilized and applied for immunofluorescent staining as described previously [34]. β-Tubulin III (TUJ1) antibody (1:200, Covance, MMS-435P, Princeton, NJ, USA) was used to detect neurons, and GFAP antibody (1:400, Dako, Z0334, Frankfurt am Main, Germany) was used to detect astrocytes.

### 2.5. Histology, TUNEL Reaction and Immunofluorescent Staining

Brains from E14.5 and E18.5 embryos were fixed overnight with 4% paraformaldehyde (PFA) (pH 7.2) and cryopreserved in 30% sucrose overnight. Neg-50 (Thermo Fischer Scientific) frozen section medium was used to embed the brains followed by cryosectioning (Microm™ HM 550 Cryostat, Thermo Fischer Scientific, Waltham, MA, USA) of 12 µm thick slices. After antigen retrieval in citrate buffer for 40 min at 95 °C, immunostaining with the following primary antibodies was performed: rabbit anti-SOX2 (1:200, Abcam, Ab97959); rabbit anti-TBR2/Eomes (1:200, Abcam, Ab23345); rabbit anti-TBR1 (1:200, Abcam, Ab31940); rat anti-CTIP2 (1:200, Abcam, Ab18465); rabbit anti-CUX1/CDP (1:100, Santa Cruz, Sc-13024, Heidelberg, Germany); rabbit anti-A-calbindin D-28k (1:1000, Swant, CB38, Burgdorf, Switzerland); anti-phospho-histone H3 (Ser10) (1:400, Cell Signaling, 9071, Danvers, MA, USA); and anti-Cleaved Caspase-3 (1:200, Cell Signaling, 9661S). Immunoreactivity was visualized using secondary antibodies anti-rabbit IgG conjugated with Cy3 (1:200, Sigma-Aldrich), Cy2 or Cy5 (1:200, Jackson ImmunoResearch), donkey anti-mouse IgG conjugated with Cy3 (1:200; Sigma-Aldrich), goat anti-rabbit Biotin conjugated (1:400, Vector Laboratories, Burlingame, CA, USA) and donkey anti-rat IgG conjugated with Alexa-488 (1:200, Sigma-Aldrich), streptavidin-Cy3 (1:800, Sigma-Aldrich). Overall cell death was detected using TUNEL reaction, as described previously [35]. In all cases, the nuclei were counterstained using DAPI (1:10,000, Sigma-Aldrich) and mounted in ProLong Gold Antifade reagent (P36930, Invitrogen).

### 2.6. Microscopy and Image Analysis

The images of whole brain sections were acquired using BX61VS Olympus Virtual microscope and processed with Olympus Olyvia 2.9 computer program. The immunofluorescence images of the in vitro differentiation of neural stem cells were acquired using the microscope ImageXpress Micro Confocal (short IXMC) from Molecular Device (MD). In each well, a z-stack of seven images with 1 µm distance was recorded using 10× Plan Apo objective with confocal mode (50 µm slit disc) at four sites. For subsequent image analysis, a custom module in the MetaXpress software from MD (version 6.2.3.) was created and applied on maximum projections of each z-stack. Each cell was defined by its nucleus using a mask derived from the DAPI channel. Depending on the intensity within this mask in the other channels, each nucleus was classified as belonging to a neuron, astrocyte or unclassified cell. This led to a summarized number of each cell type in each well.

### 2.7. RNA-Seq and Bioinformatic Analysis

NPCs were isolated from E13.5 embryo brains after crossing *Smg6*^flox^ and Rosa26-CreER^T2^ mice, and kept in neurosphere cultures for 2 days followed by 4-Hydroxytamoxifen (4-OHT, Sigma) treatment to induce the *Smg6* deletion. Total RNA was isolated 6 days after 4-OHT treatment from cells with genotypes *Smg6*^flox/flox^; Rosa26-CreER^T2^ (without 4-OHT) (ctr), *Smg6*^flox/flox^ treated with 4-OHT (ctr + 4-OHT) and *Smg6*^flox/flox^; Rosa26-CreER^T2^ treated with 4-OHT (*Smg6*-iKO)) using the RNeasy Mini Kit (Qiagen) and following the manufacturer’s manual. The RNA integrity was checked using an Agilent Bioanalyzer 2100 (Agilent Technologies). All samples showed a RIN (RNA integrity number) higher than 9. Approximately 800 ng of total RNA was used for library preparation using a TruSeq Stranded Total RNA (RiboZero Gold) according to the manufacturer’s protocol. The libraries were pooled into one and sequenced in three lanes using HiSeq2500 (Illumina) in single-read high-output mode, which created reads with a length of 50 bp. Sequencing reads were extracted using bcl2FastQ v1.8.4. On average, 73 million reads per sample were obtained. For expression analysis, the raw reads were mapped with STAR (version 2.5.4b, parameters: --alignIntronMax 100000--outSJfilterReads --outSAMmultNmax Unique --outFilterMismatchNoverLmax 0.04) [36] to the *Mus musculus* genome (GRCm38) with the Ensembl genome annotation (Release 92). For each Ensembl gene, reads that mapped uniquely to one genomic position were counted with FeatureCounts (version 1.5.0, multi-mapping or multi-overlapping reads were not counted, stranded mode was set to “–s 2”, Ensembl release 92 gene annotation) [37]. The table of raw counts per gene per sample was analysed with R (version 3.5.0) using the package DESeq2 (version 1.20.0) [38]. The sample group *Smg6*-iKO (*n* = 4) was contrasted with the sample group Ctr + 4-OHT (*n* = 4), with *Smg6*-iKO being the reference level. The sample group Ctr (*n* = 3) was also contrasted with the sample group *Smg6*-iKO (*n* = 4) and the Ctr + 4-OHT (*n* = 4). For each gene of the comparison, the *p*-value was calculated using the Wald significance test. Resulting *p*-values were adjusted for multiple testing with Benjamini & Hochberg correction. Genes with an adjusted *p*-value < 0.05 were considered differentially expressed (DEGs). The log2 fold changes (LFCs) were shrunk with lfcShrink to control for variance of LFC estimates for genes with low read counts. The changes in molecular pathways as well as their possible upstream regulators were identified by analyzing abovementioned three pairwise comparisons using Ingenuity Pathway Analysis (IPA) program (Qiagen).

### 2.8. Statistical Analysis

Depending on the distribution of the data points, unpaired two-tailed Student’s *t*-test or Mann–Whitney U (MWU) test were used to calculate significance. Data sets underwent Shapiro–Wilk test for the normal distribution. If the data set passed the Shapiro–Wilk test (*p* value > 0.05), Student’s *t*-test was used, if it did not pass (*p* value < 0.05), Mann–Whitney U test was applied. Statistical analyses were performed using GraphPad Prism 8 (GraphPad Software, San Diego, CA, USA). The type of the test performed is indicated in each figure legend. Indication for significance was used as follows: n.s. > 0.05, * < 0.05, ** < 0.01, *** < 0.001, **** < 0.0001.

## 3. Results

### 3.1. Smg6 Deletion in CNS Compromises Embryonic Neurogenesis and Newborn Viability

In order to understand the biological function of SMG6 specifically in the differentiation program of committed lineage NPCs in the central nervous system (CNS), we generated a conditional knockout mouse model in which *Smg6* was deleted in NPCs from embryonic day E10.5 by intercrossing *Smg6^F/∆^* [13] and *Nestin-Cre* transgenic mice [27] to generate CNS-deleted mice (*Smg6*-CNS∆). We obtained an expected number of *Smg6*-CNS∆ embryos at E14.5 and a slightly reduced number at E18.5 (Appendix AA) according to the Mendelian ratio. However, all *Smg6*-CNS∆ newborns died within 1–2 days after birth (Appendix AB).

To investigate the role of SMG6 in neurogenesis, we first analysed brains at E14.5 and found that an efficient *Smg6* deletion in the CNS (Appendix AC) yielded normal embryo body and brain weight as well as cortex (CTX) thickness (Appendix AD–G). Furthermore, cortical cellularity and populations of SOX2+ neural progenitor cells and TBR2+ intermediate progenitors (IPs), as well as populations of early newborn neurons positive for TBR1 and CTIP2, were the same as in controls at this stage of development (Figure 1A–C). Next, we examined the brains just before birth at E18.5 and found that *Smg6*-CNS∆ fetuses had normal body and brain weights (Appendix AH–J). However, we detected a significantly smaller CTX in E18.5 *Smg6*-CNS∆ brains (Figure 1D), indicating mild microcephaly and defects in embryonic neurogenesis. Concomitantly, the *Smg6*-CNS∆ cortices presented a significant reduction of cortical cellularity at this stage (Figure 1E), likely responsible for the reduction of the CTX.

Mouse CTX contain well defined cellular layers composed of neural precursors in the ventricular and subventricular proliferative zones (VZ and SVZ, respectively), early born neurons in the middle part (layers VI and V) and late born neurons in the upper part of the cortical plate (layers II/III). Next, we histologically analysed the structure of the E18.5 *Smg6*-CNS∆ brains. It revealed that all the neuronal layers in *Smg6*-CNS∆ cortices were formed, but with a significant decrease of SOX2+ NPCs in the VZ and TBR2+ IPs in the SVZ, indicating an exhaustion of NPC pools during the late embryonic brain development (Figure 1F,G). The numbers of early born neurons in layers VI and V positive for TBR1 and CTIP2 were normal, in contrast to a significant reduction of the late born neurons in the layers II/III judged by the CUX1+ population (Figure 1F,G). These findings indicate that SMG6 is dispensable for early cortical neurogenesis, but its deletion prematurely depleted NPC pools and compromised cortical neurogenic production, affecting the cellularity of the CTX during later development.

### 3.2. SMG6 Is Essential for Survival of Neural Cells

In order to investigate the cause for the reduction of NPCs and late born neurons of E18.5 *Smg6*-CNS∆ brains, we analysed cell death by TUNEL and Active-Caspase 3 (Act-Cas3) staining. At both E14.5 and E18.5 stages *Smg6*-CNS∆ brains exhibited a significant increase of TUNEL and Act-Cas3 signals in the areas of CTX and GE (Figure 2A–H and Appendix AA–H). The cell death at E14.5 was mainly found in the proliferative VZ and SVZ areas, where double TBR2 + TUNEL+ staining confirmed the death of IPs (Figure 2A,B). Intriguingly, TUNEL staining in GE (Figure 2C,D) detected clearly higher (Figure 2D versus Figure 2Bi) cell death in GE than in CTX at E14.5 brain sections. At E18.5, TUNEL positive cells in CTX were additionally detected in the intermediate zone (IZ) as well as in the cortical plate (CP) (Figure 2E,F), suggesting neuronal death at a late stage of brain development. However, it stayed at comparable levels to E14.5 (Figure 2Bi versus Figure 2Fi) whereas in GE (Figure 2G,H) we detected less TUNEL signals compared to younger embryos (Figure 2H versus Figure 2D). Immunofluorescence staining for Act-Cas3 confirmed elevated Caspase 3-dependent cell death in the CTX as well as in the GE (Appendix AA–H). Counting of phospho- Histone H3 (Ser10)+ cells did not reveal any difference for the number of mitotic cells in the CTX and GE between mutant and control animals at E14.5 (Appendix AA,B). Interestingly, SMG6 seems to be more crucial for the survival of NPCs of GABAergic interneurons (IN) that are generated in the GE. To confirm this, we stained E14.5 cortices with calbindin, an interneuron marker [39,40,41], and detected significantly less calbindin+ IN in *Smg6*-CNS∆ GE (Figure 2I,J) compared to controls. These observations indicate that at early neurogenesis, SMG6 absence affects the survival of NPCs prominently in the GE, and to a lesser extend in the CTX.

### 3.3. Smg6 Deletion Compromises the Self-Renewal and Differentiation Capacity of Neuroprogenitors

To investigate the renewal capacity of SMG6-deficient NPCs, we performed the in vitro neurosphere formation assay using neural stem cells isolated from the CTX or the GE at E13.5 (Appendix AA). The control and *Smg6*-CNS∆ cortical NPCs formed the same number of neurospheres after 7 days (Figure 3A,Ai), containing a comparable number of cells (Figure 3Aii), indicating a dispensable role of SMG6 in the renewal capacity of cortical NPCs. In contrast, mutant GE NPCs gave rise to a similar number of neurospheres (Figure 3B,Bi), but these neurospheres contained much fewer cells (Figure 3Bii). These results indicate that SMG6 is specifically critical for the renewal capacity of the GE NPCs, but less so for those from the CTX. Concomitantly, the mRNA expression levels of the transcription factors *Dlx1*, *Dlx2* and *Mash1,* known to drive interneuron differentiation [42,43,44,45,46,47], were dramatically reduced in the neurospheres originating from SMG6-deficient GE (Figure 3C). These findings indicate a specific role of SMG6 in interneuron progenitors.

Because SMG6 is essential for ES cell differentiation [13], we next studied the differentiation potential of *Smg6*-deleted neuroprogenitors. To this end, we used neurospheres derived originally from NPCs of the CTX or GE (Appendix AA) and induced their differentiation in vitro. We found significantly less neurons, judged by TUJ1 staining, at 6 and 8 days post differentiation (dpd) of NPCs originated from both the CTX and GE (Figure 3D,G,H,K). The percentage of GFAP-positive cells, representing astrocytes, was modestly but significantly increased in *Smg6* mutant cultures (Figure 3F,J). In addition, we observed overall reduced cellularity at 8 dpd compared to controls. It is of note that the reduction of cell numbers at 8 dpd was more prominent in differentiation cultures from the GE (76%) than in cultures from the CTX (29%) (Figure 3E,I). In summary, SMG6 plays a role in the differentiation program in vitro and is required for the survival of differentiating cells particularly of GE derived NPCs.

### 3.4. SMG6 Deficiency Exclusively in the Cortex Does Not Inhibit Corticogenesis

Nestin-Cre drives deletion of the *Smg6* gene in the whole brain; thus, CTX and GE both are affected in *Smg6*-CNS∆ mice. To further dissect the SMG6 function in different populations of NPCs, we deleted *Smg6* only in the cortical/hippocampal progenitor cells at day E9.5 by crossing *Smg6*-floxed mouse with the Emx1-Cre mouse model (designated as *Smg6*-CoHi∆). Intriguingly, in contrast to *Smg6*-CNS∆ mutants, the *Smg6*-CoHi∆ mice were born at expected ratios (Appendix AB) and were viable during the observation period of 20 months. We confirmed specific deletion of the SMG6 protein in the CTX, but not in other parts of the brain such as the GE and the hind/mid brain (Appendix AF). Notably, the SMG6 levels seem to be reduced less efficiently than in *Smg6*-CNS∆ embryos because the CTX of *Smg6*-CoHi∆ embryos contain a high number of cells, i.e., IN that are not affected by the deletion and thus have normal levels of SMG6. The body and brain weight of *Smg6*-CoHi∆ embryos as well as the CTX were normal at E18.5 (Appendix AC–E and Figure 4A,B). Moreover, the cellularity and thickness of *Smg6*-CoHi∆ cortices were the same as controls (Figure 4C–E). Microscopic analysis of immunofluorescent staining revealed a similar number of TBR1+ and CTIP2+ neurons as well as SOX2+ and TBR2+ neuroprogenitor populations between mutant and control littermates (Figure 4D,F). In contrast to *Smg6*-CNS∆, TUNEL assay did not detect obvious cell death in the CTX (Figure 5A,B) nor in the GE (Figure 5C,D). However, we detected a significant increase of cellular death only in the retrosplenial cortex of E18.5 *Smg6*-CoHi∆ embryos (Figure 5E,F), indicating that SMG6 is vital specifically and only for this small population of cortical cells.

Taken together, using two different neural specific Cre mouse models, we demonstrate that GE NPCs and interneurons are particularly vulnerable to SMG6 loss. We show that cortical NPCs and neurons are affected mainly if *Smg6* is deleted simultaneously in NPCs from both brain parts GE and CTX. We conclude that the defects of cortical neurogenesis in the *Smg6*-CNS∆ model are likely sensitized by *Smg6* deletion in the IN progenitors derived from the GE.

### 3.5. SMG6 Null Mutation Activates DNA Repair and p53 Pathways Causing Cell Cycle Dysregulation

The finding that *Smg6* deletion has a stronger effect on IN progenitors prompted us to investigate the transcriptional programs initiated by the loss of SMG6. We analysed the total transcriptome profile of SMG6-deficient NPCs isolated from E13.5 *Smg6*-CER (Smg6^flox^ crossed with Rosa26-CreER^T2^) mice brains and cultured for 6 days in the presence of 4-OHT that induces *Smg6* deletion (*Smg6*-iKO). RNA-seq data comparison of controls (Ctr + 4-OHT) with *Smg6*-iKO cells revealed 859 differentially expressed genes (DEGs) (cutoff adjusted *p* < 0.05), containing 385 upregulated and 474 downregulated DEGs (Figure 6A, Appendix A). Confirming its function in NMD, *Smg6* knockout resulted in the presence of the prominent NMD target genes within the upregulated DEGs (underlined in Figure 6A). The analysis using IPA (Ingenuity Pathway Analysis) showed that the majority of the DEGs are involved in DNA repair and cell cycle pathways (Figure 6B). Furthermore, we used the IPA upstream regulator tool to find which possible regulator was upstream, and whether the activation or silencing of it could explain the observed gene expression changes. The upstream analysis of all DEGs predicted the activation of *Trp53*, *Cdkn2a* and *Cdkn1a* genes (positive Z-score) that might have caused the detected changes (Figure 6C), indicating activation of DNA repair pathways that would lead to cell cycle arrest and eventually cell death. Interestingly, we found that *FoxM1* is among the top 30 strongest upstream regulators of DEGs, although with a negative Z-score, indicating that an inhibition of the FOXM1 function could also be a reason for the detected expression changes. FOXM1 is known to promote *Dlx1* gene expression [48] and its silencing could cause the down regulation of *Dlx1*, which we observed in neurospheres after *Smg6* deletion (Figure 3C).

## 4. Discussion

A complete deletion of key NMD genes such as *Smg1*, *Upf1* or *Upf2* results in early embryonic lethality, demonstrating an essential function of RNA metabolism/quality control during development [10,11,12]. We previously showed that *Smg6* deletion in ES cells had no impact on their viability but blocked their ability to differentiate into germ layers during mouse development. We concluded that SMG6-mediated NMD is a license factor for the cell fate determination of pluripotent stem cells [13]. Interestingly, in the present study, we show that if *Smg6* is deleted in committed neural stem cells during development at E10.5 (Nestin-Cre) or at E9.5 (Emx1-Cre), these NPCs are viable, can differentiate into various cell types and thus are able to support development of the entire brain. However, later embryonic neurogenesis and thus production of late born neurons is compromised in the Nestin-Cre mediated *Smg6* deletion model. This is in immense difference to the pluripotent ES cells, where the deletion of *Smg6* blocked differentiation through overexpression of the pluripotency gene *c-Myc* [13]. In contrast to ES cells, the overexpression of *c-Myc* during neurogenesis is known rather to support not only the self-renewal but also the neuronal differentiation of the NPCs [49,50].

Nestin-Cre mediated deletion of *Smg6* resulted in defects in both CTX and GE. In the CTX we detected a reduction of the NPC pool in VZ and SVZ and less production of late born neurons at E18.5 (Figure 1F,G), indicating that defects appear rather late in corticogenesis. Interestingly, we found that *Smg6* inactivation affects GE-derived NPCs earlier, i.e., at E14.5, judged by higher cellular death in histology and a decreased self-renewal of IN progenitors in the neurosphere assay compared to results obtained from CTX. Furthermore, we detected a reduced generation of calbindin-positive GABAergic neurons at E14.5, indicating dysregulation of the interneuron production. The striking finding is that Emx1-Cre mediated *Smg6* inactivation exclusively in cortical and hippocampal progenitors (*Smg6*-CoHi∆ mice), which spared IN progenitors, caused no comparable cortical defects. However, Emx1-Cre was reported to be more efficient and mediating stronger cortical defects than Nestin-Cre in other models [34,51,52]. *Smg6*-CoHi∆ mice were viable during the observation period of 20 months. This cell type specificity of SMG6 is supported by in vitro neurosphere assays, where we found that SMG6 null compromises only progenitors from GE but not from CTX. Thus, we conclude that SMG6 plays a less important role in the cortical progenitor renewal and differentiation, but is specifically required for IN neuroprogenitors and their derived cell lineages during brain development. Given these observations and the fact that GE-derived INs migrate tangentially to the CTX and regulate cortical neurogenesis via the GABA release [53,54], it is plausible that *Smg6* deletion caused malfunction of INs, which affected cortical neurogenesis (Figure 7).

The role of SMG6 on interneuron progenitors is reminiscent of a recent study showing that exon-junction complex factor RBM8A is critical for interneuron development [55]. Consistent with the defects of SMG6 deficient interneurons, key transcription factors that control the cell fate of interneuron progenitors were found dysregulated. *Dlx1* and *Dlx2* expression was dramatically reduced in SMG6-deficient neurospheres. DLX1 and DLX2 transcription factors promote or repress the expression of other transcription factors that are responsible for triggering IN differentiation, migration and maturation [42,43,44,45,46,47]. The downregulation of *Dlx1* gene correlates well with the IPA upstream regulator analysis of the RNA-seq data (Figure 6C), which predicted the inactivation of FOXM1, a known positive regulator of *Dlx1* [48]. Remarkably, along the same line to *Smg6*-CNS∆ mice, perinatal lethality has also been reported in mice with *Dlx1*/*2* double knock out (KO) [42,45,56] due to impaired differentiation and migration of GABAergic INs in the neocortex [45]. Disruptions of other genes such as *Gad1*, *Gad2*, *Nkx2.1* and *Sox6,* known to regulate GABAergic neuron development, were also often linked to perinatal lethality [57,58,59,60,61,62]. Unfortunately, the precise reason for the perinatal lethality of the above mentioned and *Smg6*-CNS∆ mice is unclear.

The transcriptional changes in the SMG6-deficient NPCs confirmed the defective NMD by changes in known NMD target transcripts. The majority of the top 30 dysregulated pathways belong to DNA repair, cell cycle and p53 related pathways, which can be modulated also by the NMD [63,64,65,66].

Since the transcriptome and thus the expression of NMD targets highly depends on the cell type, it is plausible that deletion of *Smg6* may have cell type-dependent consequences in cell fate. In this regard, the conditional knock-out of *Upf2* in the adult hematopoietic system preferentially compromised the viability of hematopoietic stem cells and progenitors, but not that of terminally differentiated T cells [11]. In addition, the *Upf2* null mutation did not affect the proliferation of fetal hepatocytes, but compromised their maturation process [67]. Furthermore, UPF3B was demonstrated to be very important for a subset of olfactory sensory neurons [68]. Taking these studies together, SMG6, or in general NMD, regulates cell fate programs highly depending on the cell type and developmental stage.

In conclusion, although SMG6 is essential for the differentiation of pluripotent ES cells, it is less important for the differentiation of committed cortical NPCs. Using various mouse models, we demonstrate that SMG6, as a general endonuclease of NMD for aberrant RNA, plays distinct roles in different cell types: CTX versus GE neuroprogenitor cells (Figure 7) versus ES cells. These genetic results allow predicting that the importance of NMD varies dramatically, depending on the transcriptional program of a specific cell type.

## Figures and Tables

**Figure 1 cells-10-03365-f001:**
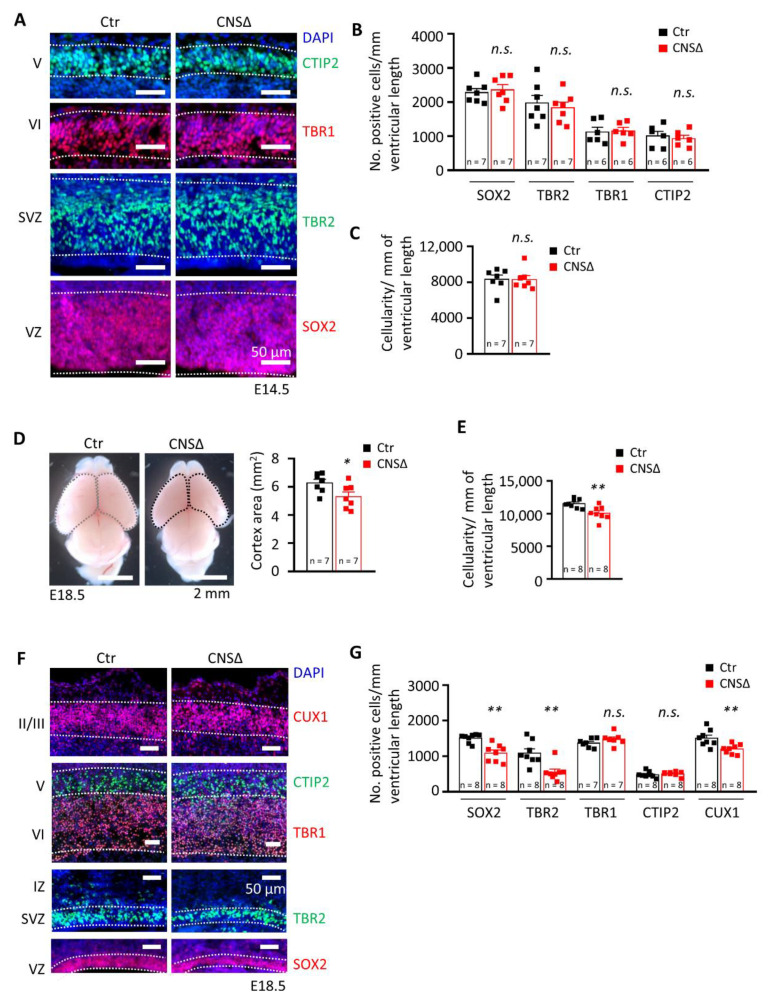
SMG6 deficiency in the central nervous system causes perinatal lethality. (**A**) Representative immunofluorescent staining of E14.5 cortices, coronal sections. SOX2 and TBR2 stained the progenitors of the VZ and SVZ, respectively. TBR1 and CTIP2 stained the early born neurons of layers VI and V, respectively. DAPI was used to stain the cell nucleus. VZ: ventricular zone, SVZ: subventricular zone. (**B**) Quantification of the neuroprogenitor cells (SOX2 and TBR2) and neurons (TBR1 and CTIP2) in E14.5 cortices in A, Student’s *t*-test was applied for all markers. (**C**) Quantification of the total cellularity in the E14.5 cortices, Student’s *t*-test. (**D**) Smaller CTX area of the E18.5 *Smg6*-CNS∆ brains, Student’s *t*-test. (**E**) Quantification of the total cellularity in the E18.5 cortices, Student’s *t*-test. (**F**) Immunofluorescent staining of E18.5 coronal sections showing only cortical region. SOX2 and TBR2 labels NPCs and intermediate progenitor cells in the VZ and SVZ, respectively. TBR1 and CTIP2 stained the early born neurons in layers VI and V. CUX1 stained the later born neurons in layers IV, III and II. DAPI was used to counterstain the cell nucleus. VZ: ventricular zone, SVZ: subventricular zone. (**G**) Quantification of the NPC cells (SOX2 and TBR2) and neurons (TBR1, CTIP2 and CUX1) in E18.5 cortices shown in D (MWU test for SOX2 and TBR2, Student’s *t*-test for the rest). For all graphs: *n*—number of embryos analysed. Error bars represent SEM, statistic comparison as indicated in each graph description—n.s. > 0.05, * *p* < 0.05, ** *p*< 0.01.

**Figure 2 cells-10-03365-f002:**
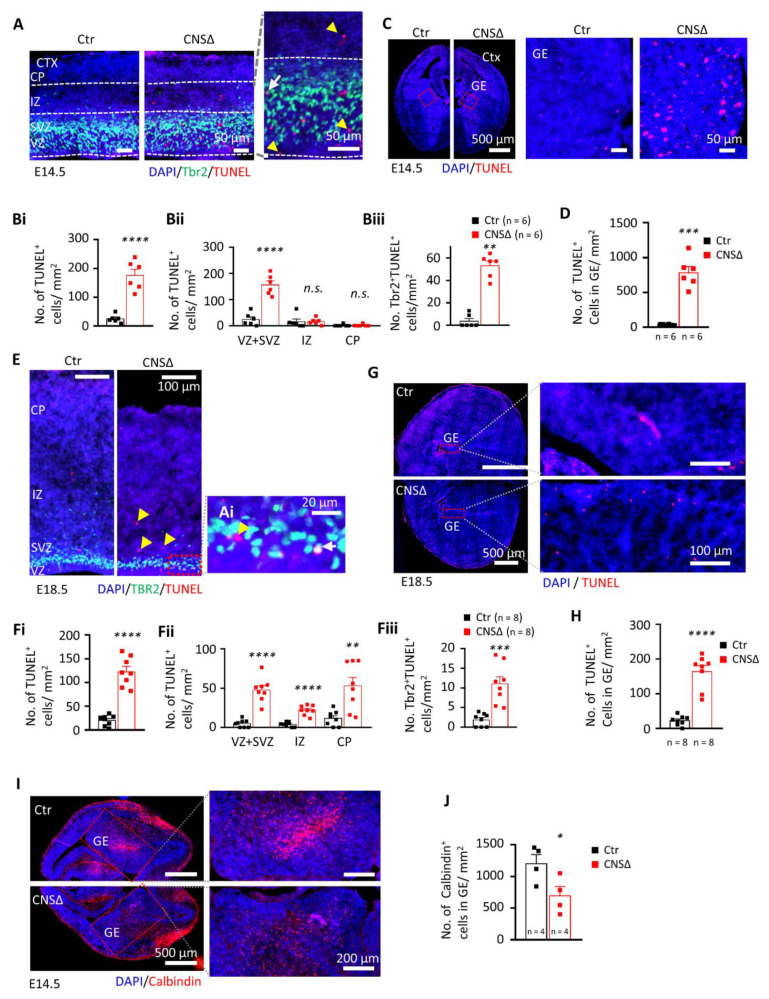
SMG6 deficiency causing death of neuronal cells. (**A**) Co-staining of dying cells using TUNEL and intermediate progenitors by labelling TBR2 on coronal sections of E14.5 embryo brains counterstained by DAPI for cell nucleus. Yellow head arrow marks TUNEL-positive cells, white arrow—TUNEL- and TBR2-positive cells. Quantification of the total TUNEL-positive cells (**Bi**) their distribution in different cortical layers (**Bii**) and dying TBR2-positive (TUNEL^+^TBR2^+^) cells (**Biii**) at E14.5 indicate neural cell death mainly in area of progenitors. Statistics MWU for IZ and CP in Bii and Biii, Welch *t*-test in Bi and Student’s *t*-test for the rest. (**C**) TUNEL staining of GE at E14.5 with quantification respectively in (**D**) using Student’s *t*-test. (**E**) Co-staining of dying cells using TUNEL and intermediate progenitors by TBR2 on coronal sections of E18.5 embryo brains counterstained by DAPI for cell nucleus. Yellow head arrow marks TUNEL-positive cells, white arrow—TUNEL- and TBR2-positive cells. Quantification performed in the same manner at E14.5 (**Fi**–**Fiii**) shows increased cell death in all cortical layers. Statistics by Student’s *t*-test. (**G**) TUNEL staining of GE at E18.5 quantified in (**H**) using Welch *t*-test. (**I**) Immuno-fluorescent staining and quantification (**J**) of calbindin-positive cells in coronal sections of E14.5 embryo brains. DAPI counterstains the cell nucleus. Statistics by Student’s *t*-test. For all graphs: VZ—ventricular zone; SVZ—subventricular zone; IZ—Intermediate zone; CP—cortical plate; *n*—number of embryos analysed. Error bars represent SEM. Significance—n.s. > 0.05, * *p* < 0.05, ** *p* < 0.01, *** *p* < 0.001, **** *p*< 0.0001.

**Figure 3 cells-10-03365-f003:**
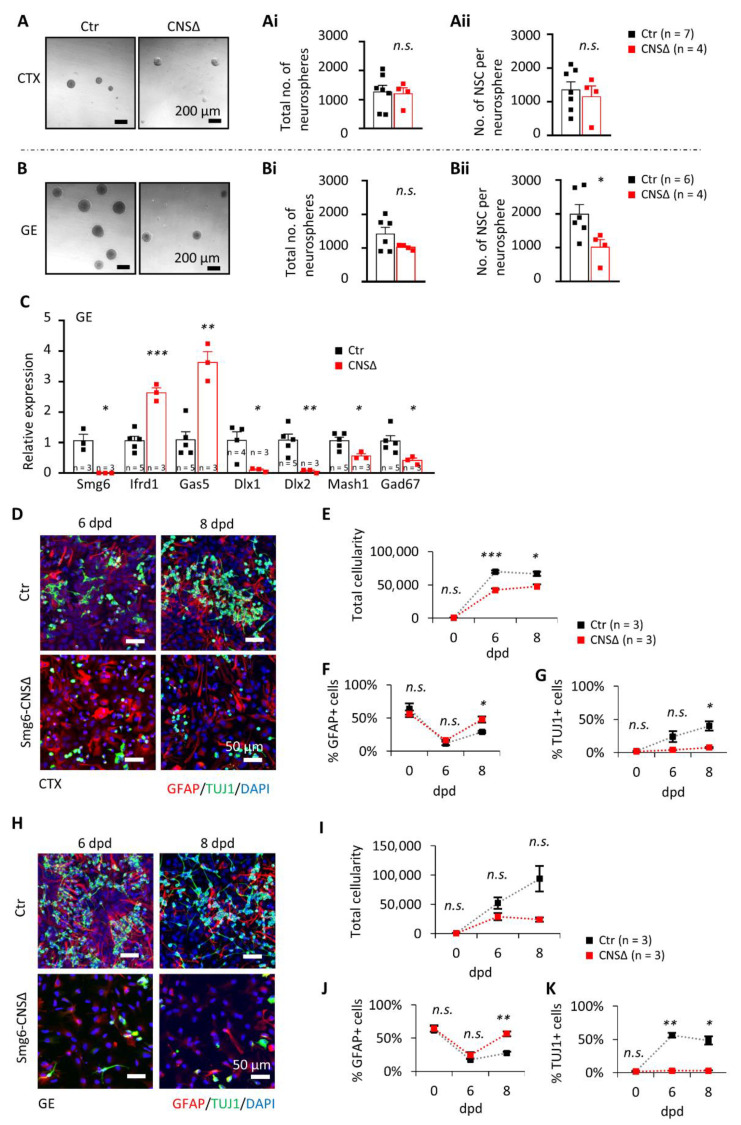
Neuroprogenitor renewal and differentiation are impaired without SMG6, especially in the NPCs from GE. (**A**) In vitro neurosphere assay on NPCs from CTX shows no significant difference in the number of neurospheres (**Ai**) as well as in the cell number per neurosphere (**Aii**) after 7 days in culture. The numbers of neurospheres formed from GE cells (**B**) also did not differ (**Bi**), but had fewer cells per neurosphere (**Bii**). (**C**) qPCR analysis on GE neurospheres shows relative expression changes of indicated gene mRNAs after *Smg6* deletion. (**D**–**G**) Differentiation capacity of progenitors from CTX and (**H**–**K**) GE at 6 and 8 days after differentiation induction (dpd). (**D**,**H**) Co-stainings of in vitro differentiated cultures from CTX and GE, respectively, at the indicated time points with quantifications of total cellularity in (**E**,**I**), GFAP-positive cells (**E**,**J**) and neurons labelled with TUJ1 staining (**G**,**K**). For all graphs: *n*—number of embryos used for cell isolations. Error bars represent SEM. Statistics by unpaired Student’s *t*-test in (**A**–**C**) and Welch *t*-test in (**E**–**G**,**I**–**K**) significance—n.s. > 0.05, * *p*< 0.05, ** *p*< 0.01, *** *p*< 0.001.

**Figure 4 cells-10-03365-f004:**
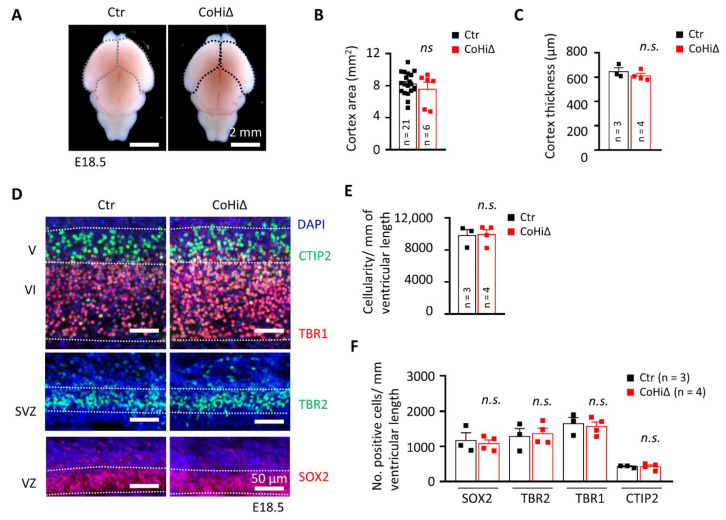
SMG6 deficiency solely in cortical and hippocampal neuroprogenitors does not inhibit corticogenesis. (**A**) Comparison of the E18.5 brains and quantification of the CTX area (**B**). (**C**) Quantification of the CTX thickness. (**D**) Immunofluorescent staining of E18.5 coronal sections showing only cortical regions. SOX2 and TBR2 labels NPCs and intermediate progenitors in the VZ and SVZ, respectively. TBR1 and CTIP2 stains early born neurons in layers VI and V. DAPI counterstains the cell nucleus. VZ: ventricular zone, SVZ: subventricular zone. (**E**) Quantification of the total cellularity in the E18.5 cortices. (**F**) Quantification of the neuroprogenitor cells (SOX2 and TBR2) and neurons (TBR1 and CTIP2) in E18.5 cortices shown in D. For all graphs: *n*—number of embryos analysed. Error bars represent SEM. Statistics by unpaired Student’s *t*-test, except in (**B**) the MWU was used, significance—n.s. > 0.05.

**Figure 5 cells-10-03365-f005:**
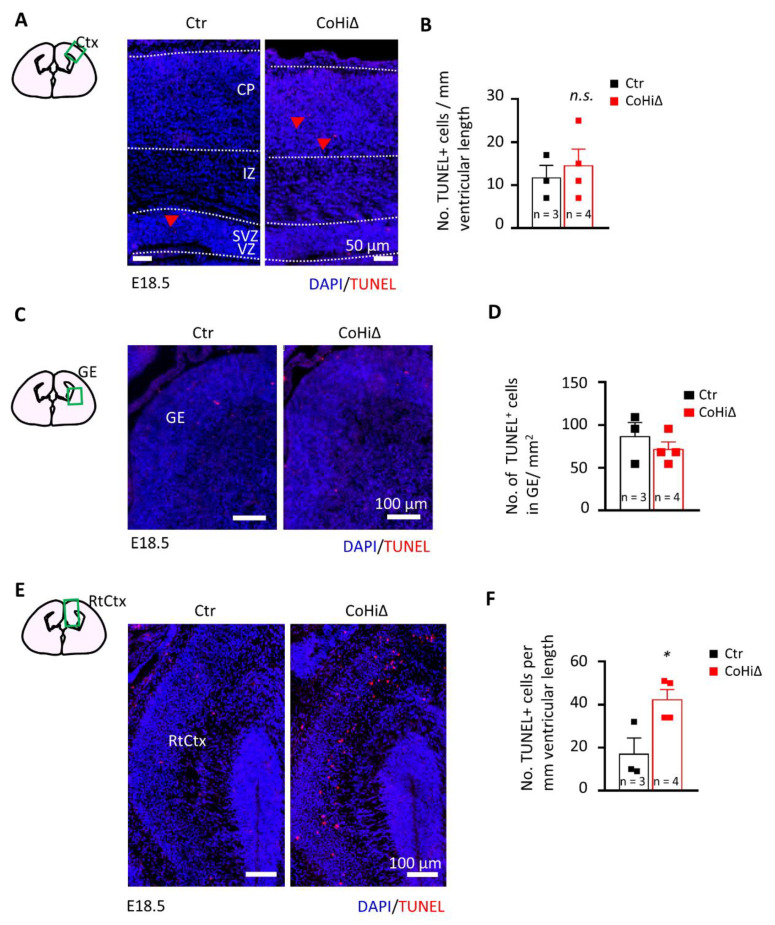
SMG6 deficiency in cortical and hippocampal neuroprogenitors cause neuronal cell death in the retrosplenial cortex. TUNEL staining comparison of E18.5 brain sections in CTX (**A**), GE (**C**) and retrosplenial cortex (RtCtx) (**D**). (**B**,**E**,**F**) Quantifications of total TUNEL positive cells in the respective brain parts. For all graphs: *n*—number of embryos analysed. Error bars represent SEM. Statistics by unpaired Student’s *t*-test, except in (**F**) where the MWU was used, significance—n.s. > 0.05, * *p*< 0.05.

**Figure 6 cells-10-03365-f006:**
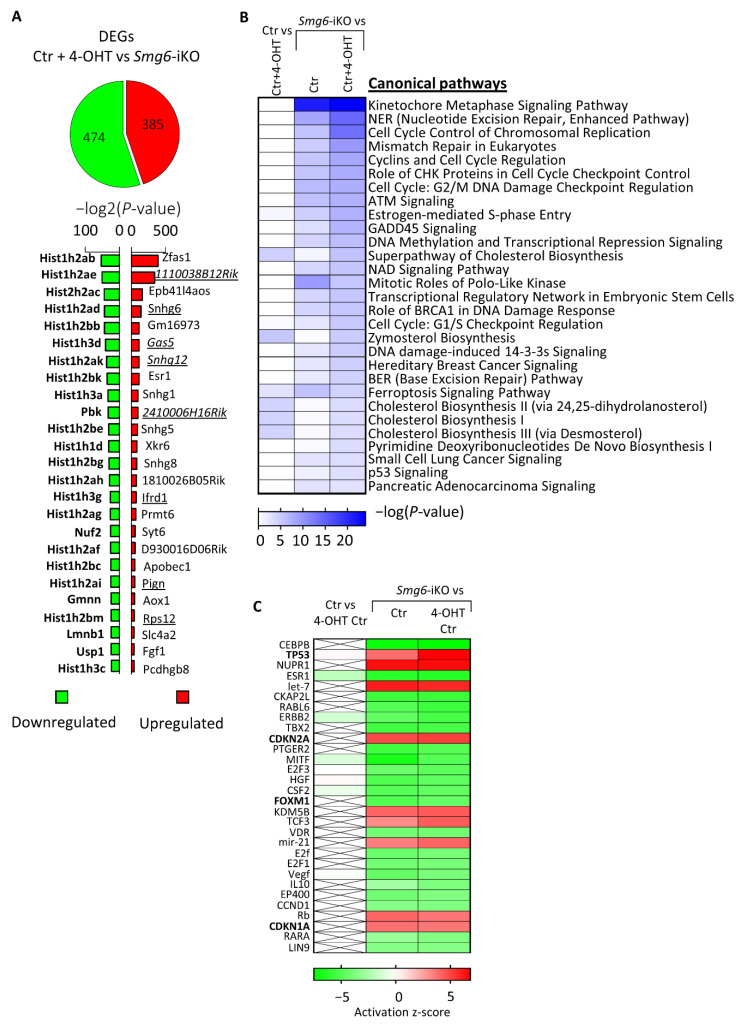
RNAseq analysis of neuroprogenitors after 4-OHT induced deletion of *Smg6*. (**A**) Total amount of identified differentially expressed genes (DEGs) with the 25 strongest up- and down-regulated genes. (**B**) Top 30 altered molecular pathways identified using IPA software. Control comparison of control cells treated and not treated with 4-OHT is shown in the left column. (**C**) Top 30 predicted upstream regulators of the detected changes in the transcriptome. Control comparison of control cells treated and not treated with 4-OHT is shown in the left column. For Ctr condition cells from 3 embryos and for each of Ctr + 4OHT and *Smg6*-iKO conditions cells from four embryos were used. DEGs for analysis by IPA were defined by adjusted *p*-value ≤ 0.05 and log2FoldChange ≤ −0.2 and ≥0.2.

**Figure 7 cells-10-03365-f007:**
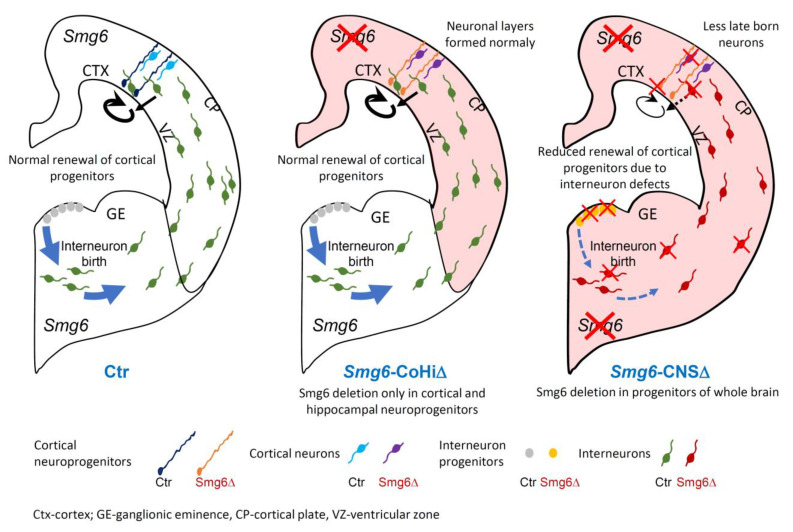
The role of SMG6 in embryonic neurogenesis. *Smg6* deletion in cortical and hippocampal NPCs (*Smg6*-CoHi∆) does not inhibit corticogenesis. However, if SMG6 is absent in the whole nervous system (*Smg6*-CNS∆) it causes cell death and also renewal defects of GE neural stem cells leading to defective interneurons tangentially migrating to the CTX. Consequently, the self-renewal of cortical NPCs and the production of late born neurons are impaired possibly because of environmental changes caused by defects in interneurons.

**Table 1 cells-10-03365-t001:** Primers used for qPCR analysis.

Transcript	Primer Sequence (5′ to 3′)	Reference
*Smg6*	AGGAATTGGACAGCCAACAG	Li et al., 2015 [13]
	TCTCGGTTTTATCCGGTTTG	
*Gas5*	TTTCCGGTCCTTCATTCTGA	Weischenfeldt et al., 2008 [11]
	TCTTCTATTTGAGCCTCCATCCA	
*Ifrd1*	ATCGGACTGTTCAACCTTTCAG	Park et al., 2017 [30]
	GCACTCTTATCAAGGGTTAGGTC	
*Gad67*	CACAGGTCACCCTCGATTTTT	MGH primerbank [31]
	ACCATCCAACGATCTCTCTCATC	
*Dlx1*	GGCTGTGTTTATGGAGTTTGGG	Wang et al., 2016 [32]
	CCTGGGTTTACGGATCTTTTTC	
*Dlx2*	GTGGCTGATATGCACTCGACC	MGH primerbank [31]
	GCTGGTTGGTGTAGTAGCTGC	
*Mash1*	TCTCCTGGGAATGGACTTTG	Kraus et al., 2013 [33]
	GGTTGGCTGTCTGGTTTGTT	
*Actin*	AGAGGGAAATCGTGCGTGAC	Li et al., 2015 [13]
	CAATAGTGATGACCTGGCCGT	
*Gapdh*	GTGTTCCTACCCCCAATGTGT	Li et al., 2015 [13]
	ATTGTCATACCAGGAAATGAGCTT	

## Data Availability

The RNAseq dataset presented in this study is available on GEO (GSE186964, https://www.ncbi.nlm.nih.gov/geo/query/acc.cgi?acc=GSE186964, accessed on 26 November 2021) and other datasets upon request from the corresponding author.

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
