# Peer review of "Cell Type-Specific Role of RNA Nuclease SMG6 in Neurogenesis"

_cells, 2021, doi:10.3390/cells10123365_

Round 1
Reviewer 1 Report
In this paper, Guerra et al. have studied the mechanisms of NMD endonuclease SMG6 in embryonic neural development. They found that when SMG6 was specifically deleted in embryonic neural stem cells using nestin-cre line (CNS∆), mice are perinatal lethal and have microcephaly. Consistently, SOX2+ and TBR2+ progenitors are reduced in the cortexes of CNS∆ mice. Interestingly, they found that SMG6 deletion led to high apoptosis in the GE and self-renewal defect in interneuron progenitors, but not in cortical progenitors. Using Emx1-cre line to knock out Smg6 in the forebrain, no dramatic phenotypes were observed, supporting that the defects of interneuron progenitor may be a major cause of phenotypes in CNS∆ mice. Lastly, RNAseq analysis has revealed some targets downstream of SMG6. These results are interesting and highly related to human neurodevelopmental diseases. The experimental design is logical and well organized. The paper is clear and easy to read. However, some major points need to be addressed before publication.
Major points
- It is not clear why authors used Smg6 F/Δ; Nestin-cre mice for Figure 1-3, but used Smg6 F/F; Emx1-Cre in Figure 4-5. Is the severe phenotypes in CNS∆ mice caused by ∆ allele or using Nestin-cre line?
- Although cell death is obvious different in control and KO mice in both the cortex and GE, will Smg6 deletion affect progenitor proliferation? Authors should use BrdU or Ki67 to test if there is any proliferation defect.
- In figure 3, authors used the neurosphere assay to measure growth. However, this in vitro assay may not reflect the in vivo condition. Authors should to directly measure proliferation using proliferation markers in KO brains. Figure 3D and 3H only measured TUJ1 and GFAP but not neuronal identity.
- Why did authors use in vitro NPC culture from Smg6 F/F; Rosa26-CreER mice for RNAseq analysis? The mice are different from Figure 1-5. Among DEGs, how many are NMD targets? Are they validated in KO mice?
- In Figure 1C, 1E and 1G, why the Y axis used cellularity/mm as the unit? It should be consistent with other figures using density as measurement.
Minor points:
- Please explain Smg6 Δ mutant in the main text.
- What are Dii and Diii in the figure 2 legend?
- Need to increase n >3 in Figure 5C-D.
- RNAseq data set should be deposited in Genebank or other public available databases. This information should be included in the methods.
Reviewer 2 Report
The reviewed study is dedicated to clarifying the role of SMG6 (Suppressor 40 with morphogenetic effect on genitalia protein 6) endonuclease that cleaves mRNA in neurogenesis during brain development in mice. The approach was to sequentially employ (a) various neural progenitor cells isolated from embryo brain regions (cortex, ganglionic eminence) of mice with inactivated SMG6 (at different stages of neurogenesis, (b) comparing the several gene expression profiles by qPCR and protein expression by TUNEL and immunofluorescent imaging analysis within the different layers in cortices and ventricular and subventricular proliferative zones.
The data demonstrate that SMG6 deficiency in early cortical neurogenesis resulted in perinatal death whereas SMG6 expression provided essential support for survival of neural cells. The novelty of the study was in determining that if Smg6 deleted in later, committed stages of neural precursors, these cells remained an ability to both self-renewal and differentiation to develop brain structures. The strength of the study is that it clearly demonstrates that depending on cell progenitors stage the SMG6 expression might distinctly affect the fate of cortex versus ganglionic eminence committed progenitors.
Despite intensive explanation in discussion, however, the text requires an additional figure/scheme (for the discussion part) to illustrate the details of the role of SMG6 expression in neural precursor cells fate depending on a stage and migration capabilities from ganglionic eminence region to the cortex etc., along with indicating the SMG6 expression requirements.
Author Response
Response to Reviewer 2 Comments
We thank for the positive view on the manuscript.
Despite intensive explanation in discussion, however, the text requires an additional figure/scheme (for the discussion part) to illustrate the details of the role of SMG6 expression in neural precursor cells fate depending on a stage and migration capabilities from ganglionic eminence region to the cortex etc., along with indicating the SMG6 expression requirements.
Response: We had assembled a working model to explain our main findings and our thoughts in the original Supplementary Fig. S4. Following this reviewer’s suggestion, we revised the working model and now moved it to the main figure (Rev Fig 7) in order to help readers to follow the discussion of our study.
Round 2
Reviewer 1 Report
The revision addressed my major concerns.